# Skin Testing and Basophil Activation Testing Is Useful for Assessing Immediate Reactions to Polyethylene Glycol-Containing Vaccines

**DOI:** 10.3390/vaccines11020252

**Published:** 2023-01-23

**Authors:** Jamma Li, Christopher Weir, Richard Fulton, Suran L. Fernando

**Affiliations:** 1Department of Clinical Immunology and Allergy, Royal North Shore Hospital, Sydney 2065, Australia; 2Faculty of Medicine and Health, University of Sydney, Sydney 2050, Australia; 3New South Wales Health Pathology, Sydney 2000, Australia; 4Northern Blood Research Centre, University of Sydney, Sydney 2065, Australia

**Keywords:** allergy, drug allergy, vaccine allergy, polyethylene glycol, COVID-19, COVID-19 vaccine

## Abstract

Background: The mechanism of immediate reactions to drugs or vaccines containing polyethylene glycol (PEG) and PEG derivatives is not fully elucidated. It is considered in many instances to be IgE-mediated. Diagnosis and management of PEG allergy is topical, as BNT162b and mRNA-1273 contain PEG (2[PEG-2000]-N), and ChAdOx1-S and NVX-CoV2373 contain polysorbate 80. mRNA vaccines contain PEG 2000, which encapsulates the mRNA to impair its degradation. This PEG MW is specific to mRNA vaccines and is not used in other drugs and vaccines. PEG 2000 allergy is not well studied, as higher PEG molecular weights are implicated in most of the PEG allergy published in the literature. Methods: We performed a literature review on PEG allergy and sought to evaluate the safety and effectiveness of our protocol for assessment of PEG 2000 and polysorbate 80 reactions in an outpatient clinic setting. All patients referred to our drug allergy service between 1 July 2021 and 31 December 2021 with suspected immediate allergy to PEG or its derivatives were eligible for the study. Skin testing (ST) and basophil activation testing (BAT) were performed for all patients to multiple PEG molecular weights (MWs). Results: We reviewed twenty patients during the study period. Five patients were allergic. Fifteen patients had a masquerade of allergy and were enrolled as control patients. PEG 2000, polysorbate 80, BNT162b, and ChAdOx1-S had excellent performance characteristics on skin testing. BAT showed high specificity for all vaccines and PEG MWs. Discussion: In our small study, we found ST and BAT to add useful information, particularly for PEG 2000 allergy. Further study of our protocol in larger patient cohorts will provide more information on its performance characteristics and usefulness.

## 1. Introduction

Anaphylaxis to vaccination is a serious adverse event (SAE) following immunization. Although vaccine anaphylaxis is rare, it occurs mainly in those who have allergy to a vaccine component [1]. Anaphylaxis occurs rapidly and systemically, usually in systems in which there are large numbers of mast cells [2]. The culprit allergen engages with a receptor on the mast cell, resulting in mast cell degranulation. Clinical manifestations range from non-life-threatening derangements in one or more major organ systems, to life threatening cardiovascular or respiratory derangements, and cardiac or respiratory arrest. Anaphylaxis may be described as immunoglobulin E (IgE)-mediated anaphylaxis, in which the culprit allergen binds to the high affinity IgE receptor, or non-IgE mediated anaphylaxis, in which it engages with any other receptor, for example, complement receptors.

The Centers for Disease Control and Prevention (CDC) in the United States reported 11.1 cases of anaphylaxis per million first doses of BNT162b2 administered [3], and 2.5 cases per million first doses of mRNA-1273 administered [4]. More recently, the overall anaphylaxis rate for licensed COVID-19 vaccines (BNT162b2, mRNA-1273, ChAdOx1-S, and Ad26.COV2-S) in the United States and Europe was calculated at 10.67 per million doses. The incidence of anaphylaxis for COVID-19 vaccines, therefore, is higher than that for vaccination generally prior to COVID-19; the 2016 Vaccine Safety Datalink study reported an overall incidence of 1.31 cases per million vaccine doses [5,6].

In Australia, licensed mRNA COVID-19 vaccines comprise BNT162b2 and mRNA-1273, which contain polyethylene glycol (PEG) 2000. PEGs are a group of polyether compounds which are widely used in medicines, cosmetics, and household products and as a food additive and in various industrial processes. Polymerization of ethylene oxide results in PEG polymers of variable length and molecular weight (MW) [7]. PEG may be conjugated to different molecules. BNT162b2 contains PEG 2000 in the form of 2[PEG-2000]-N, and mRNA-1273 contains PEG 2000 in the form of PEG2000-DMG. PEG is also found in PEG derivatives, including PEG ethers, PEG fatty acid esters, PEG castor oils, PEG-propylene glycol copolymers, and PEG soy sterols. Most routine exposures to PEG that occur are to MWs under 75 g/mol, such as that in toothpaste, cleansers, cosmetics, and creams, as well as to 3350 g/mol (PEG 3350) in osmotic laxatives and 400 g/mol (PEG 400) and 6000 g/mol (PEG 6000) in drug excipients. Low MW (<400 g/mol) PEGs penetrate skin and mucosa more readily and increases risk of sensitization.

PEG is increasingly used as an excipient in drugs to prolong the circulation time by impeding metabolism or degradation, as effective pill binders, the active ingredient in osmotic laxatives, stabilizers in suppositories, bone cement, lubricants, gels, and liquids for parenteral administration. For COVID-19 mRNA vaccines, PEG 2000 is crucial for the formation of a pegylated nanoparticle that encapsulates the mRNA, impairing its degradation and increasing its water solubility and, ultimately, the bioavailability of the lipid nanoparticle [8].

ChAdOx1-S and NVX-CoV2373 contains polyoxyethylene-sorbitan-20-monooleate (also known as polysorbate 80 and Tween 80), an emulsifier and solubilizing agent ubiquitously used in foods (E433), creams, ointments, lotions, tablets, anticancer agents, and vaccines including DTaP, Hep B, HPV, pneumococcal, influenza, and herpes zoster [9]. Ad26.COV2-S, which is widely administered in the USA, Europe, South Africa and Brazil, also contains polysorbate 80.

PEG is a proven allergenic component of vaccines, and polysorbate 80 is a suspected allergenic component [1]. Polysorbates are obtained from PEG moieties but have lower molecular weights and may be less likely to induce hypersensitivity. For example, polysorbate 80 has a molecular weight of 1310 Da [10]. IgE sensitization to polysorbate 80 (without allergy) is more common than allergy to polysorbate 80, which may explain the tolerance of PEG-allergic patients to polysorbates in substances such as food [11].

PEG allergy is uncommon, with most reactions reported being caused by high MW PEGs. At present, PEG 3350 and PEG 4000 account for most described cases of IgE-mediated PEG anaphylaxis in the literature, approximately 55%, followed by PEG 6000, which accounts for about 20% [11]. In most cases, exposure is oral, but intravenous, intramuscular, intra-articular, and topical exposure is also described. Allergy symptoms typically occur within minutes of exposure. PEG anaphylaxis is likely to be dose-dependent; each PEG-allergic patient may have an individual threshold level, which is dependent on both the MW and amount of PEG administered [12]. Reactivity to multiple MW is well described, but the nature of this cross-reactivity is not well understood [13,14]. Reactivity across low and high MWs (>4000 g/mol) or to high MWs only are described with reports of PEG-allergic subjects having an individual threshold level dependent on MW in combination with the amount of PEG ingested. Furthermore, cross-reactivity may occur between PEG and PEG derivatives [11].

The mechanism of anaphylaxis to COVID-19 vaccines containing PEG and PEG derivatives is not fully elucidated. The role of IgE-mediated mechanisms remains unclear. Earlier reports during the COVID-19 pandemic showed positive skin test responses to higher MW PEG (PEG 3350), suggesting a role for its use in predictive skin testing in those at risk of PEG allergy, and as confirmatory testing following COVID-19 vaccine anaphylaxis [15,16]. The identification of pre-existing, anti-PEG IgE in 2 of 2091 normal sera by double bead cytometry provides a potential explanation for reactivity on first exposure although it is unclear whether these antibodies trigger mast cell activation [17].

However, non-IgE mechanisms may be implicated in COVID-19 vaccine anaphylaxis, as most IgE-mediated reactions in the literature are to PEG with MW>3350 [18]. Warren et al. demonstrated IgG rather than IgE-antibodies to PEG in 11 patients with anaphylaxis to BNT162b2, possibly accounting for the positive BAT and negative ST results and implicating complement-activation-related pseudoallergy (CARPA) [19]. This mechanism may account for cases of successful subsequent administration of the vaccine in incremental doses and pretreatment with antihistamines [20,21].

Furthermore, non-allergic presentations—for example, immunization stress-related responses (ISRR) [22,23] and infusion reactions [22]—may have symptoms that overlap with anaphylaxis. ISRR comprise acute stress response, vasovagal reaction, and dissociative neurological symptom reactions [22,23]. Infusion reactions may be reactions to foreign proteins contained in the infusion or non-immune related reactions, e.g., through cytokine release, as in cytokine release syndrome [24]. There is significant overlap in skin, cardiovascular, respiratory, gastroenterological, and neurological symptoms between ISRR, infusion reactions, and anaphylaxis (Table 1). Consequently, ISRR and infusion reactions may be mistaken for anaphylaxis.

Skin testing (ST), comprising skin prick testing (SPT), intradermal testing (IDT), and basophil activation testing (BAT) are used to aid diagnosis of drug and vaccine anaphylaxis including PEG anaphylaxis. ST generally comprises SPT at first instance, as systemic reactions from SPT are well described in PEG allergic patients; however, escalation to IDT is possible after negative SPT [11]. ST with higher PEG MW is more likely to result in systemic reaction in allergic patients [11]. Recommendations have continued to evolve, and there remains no standardized method. In 2020, shortly after COVID-19 vaccinations became available, Banerji et al. provided guidance in patients with allergy or atopy for first-time COVID-19 vaccine administration [25]. The authors suggested ST for high-risk patients, comprising those with suspected PEG anaphylaxis. ST was performed as SPT to PEG 3350-, polysorbate 80-, and polysorbate 20-containing medications and, if negative, followed by IDT [25]. In 2021, Ieven et al. identified PEG allergic patients, predominantly to PEG 3500/4000 but also to PEG 400 and PEG 6000, who were able to tolerate polysorbate 80 and polysorbate 80-containing vaccines by ST [26]. This was performed as a SPT to a range of PEG MW in pure form and PEG-containing medications and to polysorbate 80 in pure, undiluted form [26]. IDT was performed only to PEG 3350 and polysorbate 80 if SPT was negative [26]. In 2022, the European Association of Allergy & Immunology (EAACI) identified high-risk groups for COVID-19 vaccination, comprising those with immediate reactions (within 2 h of administration) to COVID-19 vaccines, or anaphylaxis; and those with a history of suspected PEG anaphylaxis [27]. It was recommended that these patients are referred for allergy assessment with SPT to COVID-19 vaccine in neat concentration, and PEG and polysorbate 80 in either pure form at up to 50% dilution in water, or as PEG-containing medications [27]. Further guidance for SPT testing concentrations to pure PEG was described by Bruusgaard-Mouritsen et al. in 2022. The authors suggested using PEG 300 at 100% weight/volume (*w*/*v*), PEG 3000 at 50% *w*/*v*, PEG 6000 at 50% *w*/*v*, and polysorbate 80 at 20% *w*/*v* [28]. SPT to PEG 20,000 was recommended to be performed in a stepwise fashion with 0.01%, 0.1%, 1%, 10%, and 20% *w*/*v* until a positive response was reached. The protocol, however, did not include PEG 2000. In addition to testing the culprit PEG MW, it was considered that ST with an MW higher than the culprit MW may improve ST sensitivity, and should be performed prior to excluding PEG allergy [11].

Basophils are found in peripheral blood and serve as a model for human mast cells. In BAT, basophil activation occurs after an allergen, e.g., PEG, binds to a receptor on the resting basophil. On the resting basophil, there is no constitutive expression of the basophil surface marker CD63, although there is some constitutive expression of CD203c. Activation of basophils triggers upregulation of these basophil surface markers. There is no standardized methodology for BAT for the evaluation of immediate reactions to COVID-19 vaccines, and there is variance in the threshold for positivity [29]. As mentioned, BAT does not necessarily distinguish between IgE and non-IgE-mediated hypersensitivity. Furthermore, in Australia and most parts of the world, this is not a diagnostic laboratory test and available only as a research assay.

## 2. Materials and Methods

In light of the clinical need to optimize assessment of PEG anaphylaxis, we aimed to study the usefulness of ST and BAT for the assessment of PEG allergy. We focused on the assessment of PEG 2000 and polysorbate 80 allergy and safe subsequent administration of the COVID-19 vaccine in an outpatient clinic setting. We wanted to use only COVID-19 vaccines and PEG in pure form for both ST and BAT.

Patients referred to our teaching hospital drug allergy service between July 1 and 31 December 2021 with suspected immediate anaphylaxis to PEG or its derivatives (within 4 h of administration) were eligible for this study. Informed consent was obtained from all participants. All patient reviews and drug allergy testing were performed in a standard outpatient clinic setting, in clinic rooms with an examination bed and access to oxygen and suction. Patients were classified into allergic and non-allergic groups for each PEG MW. Patients with history and examination consistent with anaphylaxis for a PEG MW were classified as allergic. We used the Brighton Collaboration case definition of anaphylaxis to guide our classification [30]. Cases of Level 1, 2, or 3 of certainty were included as allergic. Patients who tolerated or had symptoms not consistent with anaphylaxis to PEG and BNT162b and/or ChAdOx1-S were classified as non-allergic. These non-allergic symptoms are classified as ISSR and infusion reactions.

All patients received ST to the culprit drug/vaccine, and PEG 400, 1500, 2050, 3350, 6000, and polysorbate 80 (Sigma-Aldrich, St-Louis, MI, United States). This represents all the PEG between 400 g/mol and 6000 g/mol inclusive that we could purchase. PEG with MW above 6000 g/mol was not utilized, as the outpatient clinic risk matrix did not permit use due to the high risk of systemic reaction. ST to PEG 400 was performed as SPT to neat, and intradermal testing (IDT) was performed at serial dilutions from 0.01% *w*/*v* to neat in sterile water. ST to other MW was performed as SPT to 50% *w*/*v*, and IDT was performed to serial dilutions from 0.005% to 50% *w*/*v*. ST to BNT162b and ChAdOx1-S was performed as SPT to 10% *w*/*v*, and IDT was performed at 10% and 1% *w*/*v*.

BAT was performed to culprit and tolerated PEG MW, and vaccine was performed in line with our protocol [31]. Serial dilutions of 0.1% to 10% *w*/*v* were used for all PEG, and serial dilutions of 2.5% to 10% *w*/*v* were used for BNT162b and ChAdOx1-S. Upregulation of 5% for CD63 and 20% for CD203c was used as the cut-off for a positive result.

## 3. Results

Twenty patients were enrolled. The process of assessment and results for these 20 patients are shown in Figure 1. Five patients had IgE-mediated allergy. Two patients had anaphylaxis to both paclitaxel and BNT162b, two to BNT162b alone, and one to ChAdOx-S. All patients developed allergic symptoms within 20 min of exposure. All patients had positive ST to the drug/vaccine culprit and culprit PEG/polysorbate MW. Patient characteristics and testing results are shown in Table 2.

Fifteen patients were found to have non-allergic presentations and were used as controls. The median age of these controls was 46 years. Thirteen of fifteen patients were female. Two of fifteen patients reported a history of atopy. Five patients were unvaccinated against COVID-19 at the time of review; four patients had infusion reactions to paclitaxel, which responded to slowing of the infusion rate. One patient was referred for assessment of PEG allergy, but scrutiny of her current regular medications demonstrated tolerance of multiple PEG MW. Ten patients received dose 1 of a COVID-19 vaccination; eight patients had ISRR to BNT162b and 2 to ChAdOx-S.

For both allergic and non-allergic patients, individual ST data were matched with each patient’s clinical history of allergy and tolerance to PEG MW. Results for PEG 1500 and PEG 2050 were identical. The performance characteristics of using various concentrations as a cut-off for positivity are summarized in Table 2. All SPT showed negative sensitization, so only IDT results are included. For BAT, cut-offs for up-regulation of 5% and 20% for CD63 and CD203c, respectively, were used. Performance characteristics at specified concentrations for ST and BAT are shown in Table 3. For PEG 400 and 3350, we show specificity only at serial concentrations, as we did not have allergic patients. There was no significant difference in performance characteristics using the 5 unvaccinated controls compared to all 15 vaccinated and unvaccinated controls in the calculation.

All five allergic patients were subsequently vaccinated with an alternative COVID-19 vaccine with no adverse reaction (Table 2). Challenge data were additionally collected to other PEG MW derivatives to determine cross-reactivity in these patients. All 15 non-allergic patients subsequently tolerated BNT162b in our clinic, which, at the time of repeat vaccination, was the recommended first-line COVID-19 vaccine.

## 4. Discussion

Although our study comprising five allergic patients is small, true BNT162b and ChAdOx-S anaphylaxis is rare. In five allergic patients, we found ST and BAT using COVID-19 vaccines and pure PEG to add useful information, particularly for PEG 2000 allergy. Adding to the existing literature [25,26,27], we provide information on the safety of IDT using pure PEG at specified concentrations and provide some guidance on irritant concentrations for IDT. We provide an outpatient-friendly protocol using vaccines and pure PEG only to simplify ST. We found ST to PEG 1500/2050, polysorbate 80 and both BNT162b and ChAdOx-S vaccines to have excellent performance characteristics in five patients with anaphylaxis at the concentrations tested. The difference might be explained by our use of IDT, which used pure PEG of MW of derivatives found in the vaccines, and specialist assessment to separate IgE-mediated allergy from ISRR.

As ST is mostly performed as SPT, we wanted to provide our findings on the rate of IDT sensitization in tolerant patients. Indeed, there was significant sensitization on IDT to PEG 400, 3350, and 6000 in tolerant patients. To achieve a satisfactory false positive rate, IDT concentrations of PEG 400 at 1% *w*/*v*, and PEG 3350, 6000 at 0.5% *w*/*v* were desirable from our minimal data; however, there would be an impact on sensitivity. We were not able to establish other performance characteristics due to a lack of patients allergic to these MWs.

Our study population had PEG 2000 allergy mostly, as this is the MW found in the BNT162b and mRNA-1273 vaccines. However, higher MW PEG allergy (PEG 3350, 4000, or 6000) predominates in the literature on PEG allergy due to the widespread use of high MW PEG as effective osmotic laxatives and solvents in medical preparations, and the speculation that lower MW PEGs are unable to cross link specific IgE. PEG 2000 excipients have been specifically introduced for mRNA vaccines and may be considered relatively high MW capable of cross linking IgE. Individuals may develop specific IgE to PEG 2000 due to limited routine exposure to PEG 2000 compared to more ubiquitous exposure to other MWs or may develop cross-sensitization to other high MW PEGs [11].

In contrast to other groups, we found limited utility in skin prick testing alone [25,26,27,28,32]. We did not experience life-threatening reactions to any ST in our study. Two patients with systemic reactions to PEG 1500/2050 IDT had symptoms, which resolved with antihistamines.

It is proposed that non-IgE mediated mechanisms are predominantly implicated in anaphylaxis to COVID-19 vaccinations, especially CARPA due to the binding of preexisting anti-PEG IgM to the liposomes with subsequent complement activation [19]. In addition to IgE antibodies to PEG in normal sera, anti-IgG and anti-IgM were also found in greater prevalence, again accounting for reactions on first exposure [17]. It is plausible that more than one mechanism may account for immediate hypersensitivity reactions to COVID-19 reactions. We propose our ST algorithm for determination of the presence of IgE-mediated anaphylaxis using the implicated vaccines, the specific excipients (PEG 1500/2050), and polysorbate 80 at appropriate concentrations. Testing of other MW PEGs is determined by clinical history. A negative ST does not exclude anaphylaxis, as non-IgE mechanisms such as CAPRA may be implicated. Administration of future vaccines with negative ST warrants consideration of precautions such as administration of antihistamines or incremental or staged doses of the vaccine [33].

The limitation of the study is the small sample size requiring further evaluation in a larger multi-center clinical validation. We would also further investigate our skin testing results with specific IgE and preferably a radio allegro sorbent (RAST) inhibition immunoassay using the vaccines and PEGs of various MWs [34]. We attempted correlation with BAT but found the results to be mixed in our cohort. This is consistent with the minimal data available in the literature [29]. Sensitivity ranges from 0–100%, and the threshold for basophil activation used ranges from ≥4% to ≥25%. Primarily CD63 was used as a marker of basophil activation, although one study using CD203c reported a sensitivity of 100% for one patient. In our study, we found CD63 to have superior performance characteristics to CD203c. We found high specificity using BAT but low sensitivity. We found no significant difference using pure PEG compared to BNT162b in contrast to a recent report [35]. For our practice, the high cost of BAT would limit its feasibility of routine use, as its role based on our findings would primarily be for exclusion of allergy. We did not challenge our ST positive patients to another mRNA vaccine given the perceived clinical risk and the alternatives available.

## 5. Conclusions

Anaphylaxis to COVID-19 vaccination is a rare but serious SAE following vaccination. The mechanism of anaphylaxis may be IgE as well as non-IgE mediated. PEG is a proven allergenic vaccine component and may be responsible for a significant proportion of vaccine reactions. mRNA vaccines contain PEG 2000, which encapsulates the mRNA to impair its degradation. This PEG MW is specific to mRNA vaccines and is not used in other drugs and vaccines, and there is limited exposure otherwise to PEG 2000 in the community.

Skin testing and BAT are available modalities for assessment of patients with a convincing history of anaphylaxis to a COVID-19 vaccine or PEG. In our study, ST provided a suitable, practical, and less labor-intensive alternative to BAT and may indicate patients with IgE-mediated cause for COVID anaphylaxis from their specific excipients. ST and BAT may help clinicians determine the next steps in guiding patients on their choice of COVID-19 vaccine as well as precautions in its administration. Further study of our ST protocol using vaccine and pure PEG only in larger patient cohorts will provide more information on its performance characteristics and usefulness.

## Figures and Tables

**Figure 1 vaccines-11-00252-f001:**
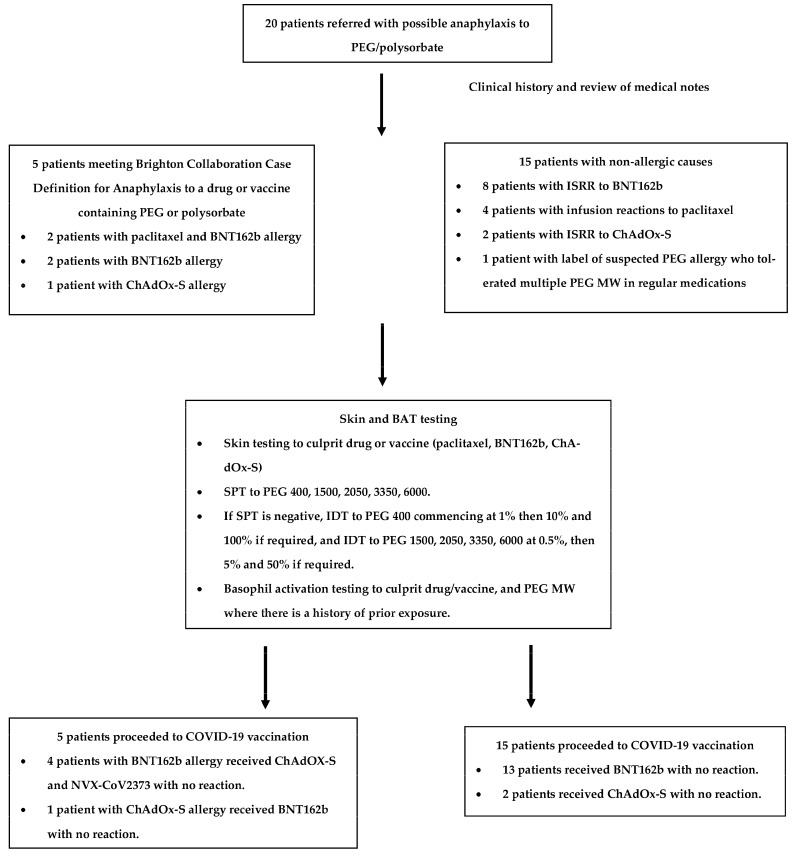
Process of assessment and results for enrolled patients.

**Table 1 vaccines-11-00252-t001:** Symptoms of anaphylaxis, immunization stress-related responses, and infusion reactions.

	Anaphylaxis	Immunization Stress-Related Responses		
		Acute stress response	Vasovagal reaction	Infusion reaction, including the cytokine release syndrome
Skin	Urticaria Erythema Pruritus Angioedema Rhinoconjunctivitis	Pallor Diaphoresis Cold and clammy	Pallor Diaphoresis Cold and clammy	Erythema Urticaria Pruritus
Cardiovascular	Tachycardia Hypotension Cardiac arrest	Tachycardia Hypertension	Bradycardia Hypotension	Hypotension
Respiratory	Cough Stridor Wheeze Respiratory arrest	Hyperventilation	Normal to deep breaths	Wheeze
Gastrointestinal	Nausea Vomiting Abdominal cramping	Nausea	Nausea Vomiting	Nausea Vomiting Abdominal cramping Diarrhea
Neurological	Uneasiness Restlessness Agitation Loss of consciousness with no response to supine position	Fearfulness Light-headedness Dizziness Paraesthesia Spasms of hands and/or feet	Transient loss of consciousness with good response to supine position	

**Table 2 vaccines-11-00252-t002:** Clinical Characteristics and Results of Intradermal Testing and Basophil Activation Testing in Allergic Patients.

Patient	Age (Years)	Gender (M/F)	Culprit Drug	Atopy	Brighton Collaboration Case Definition for Anaphylaxis [30]	Time from Reaction to Assessment	Intradermal Skin Test Results	BAT Results	History of Exposure to PEG	Outcome of COVID-19 Vaccination
1	58	F	Paclitaxel (containing PEG 35-castor oil)	nil	Level 1 Anaphylaxis Sudden onset and rapid progression of Major Criteria: - Generalized pruritus with rash, and - Measured hypotension	1 year, 4 months	Paclitaxel 0.12 mg/mL +ve	Paclitaxel -ve	n/p	
							BNT162b 10% +ve	BNT162b CD63 -ve CD203c +ve		BNT162b allergy: Systemic reaction to BNT162b 10% skin test characterized by generalized itch and rash.
							PEG 2050 50% +ve 5% +ve	PEG 2050 CD63 +ve CD203c -ve		
							ChAdOx-S 10% -ve	ChAdOx-S -ve		ChAdOx-S tolerated.
							Polysorbate 80 20% -ve	n/p		
							PEG 400 100% +ve 10% -ve	n/p	Patient is not aware of exposures to PEG 400.	
							PEG 3350 5% +ve 0.5% -ve	PEG 3350 -ve	Movicol and Coloxyl as laxative tolerated.	
							PEG 6000 5% +ve 0.5% -ve	n/p	Patient is not aware of exposures to PEG 6000.	
2	38	F	Paclitaxel (containing PEG 35-castor oil)	nil	Level 2 Anaphylaxis Sudden onset and rapid progression of: Major criteria - Generalized pruritus with rash, and - Tachypnoea Minor Criteria - Difficulty breathing without wheeze or stridor - Abdominal pain	4 year, 8 months	Paclitaxel 0.0012 mg/mL +ve	Paclitaxel 0.012 mg/mL CD63 +ve CD203c -ve		
							BNT162b 10% +ve	BNT162b CD63 +ve CD203c +ve		BNT162b allergy: Systemic reaction to BNT162b 10% skin test characterized by generalized itch and rash.
							PEG 2050 50% +ve 5% +ve	PEG 2050 CD63 +ve CD203c +ve		
							ChAdOx-S 10% -ve	ChAdOx-S CD63 -ve CD203c +ve		ChAdOx-S tolerated.
							Polysorbate 80 20% -ve	Polysorbate 80 CD63 -ve CD203c -ve		
							PEG 400 100% +ve 10% +ve 1% -ve	n/p	Patient is not aware of exposures to PEG 400.	
							PEG 3350 50% -ve	PEG 3350 -ve	Movicol as laxative tolerated.	
							PEG 6000 5% +ve 0.5% -ve	n/p	Patient is not aware of exposures to PEG 6000.	
3	47	F	BNT162b	nil	Level 1 Anaphylaxis Sudden onset and rapid progression of Major criteria: - Generalized pruritus with rash, and localized angioedema (facial) - Upper airway swelling (throat, uvula, and larynx)	8 weeks	BNT162b 10% +	BNT162b -ve		BNT162b anaphylaxis.
							PEG 2050 50% +ve 5% +ve	PEG 2050 -ve		
							ChAdOx-S 10% -ve	n/p		ChAdOx-S and NVX-CoV2373 tolerated.
							Polysorbate 80 20% -ve	n/p		
							PEG 400 100% +ve 10% +ve 1% -ve	n/p	Patient is not aware of exposures to PEG 400.	
							PEG 3350 50% +ve 5% +ve 0.5% -ve	n/p	Possible use of laxatives previously with no reaction.	
							PEG 6000 50% +ve 5% -ve	n/p	Patient is not aware of exposures to PEG 6000.	
4	34	F	BNT162b	nil	Level 2 Anaphylaxis Sudden onset and rapid progression of Major criteria: - Upper airway swelling (throat and uvula), tachypnoea, and increased use of accessory muscles Minor criteria: - Generalized prickle sensation - Difficulty breathing without wheeze or stridor	8 weeks	BNT162b 10% +ve	BNT162b -ve		BNT162b anaphylaxis.
							PEG 2050 50% +ve 5% +ve	PEG 2050 -ve		
							ChAdOx-S 10% -ve	n/p		ChAdOx-S and NVX-CoV2373 tolerated.
							Polysorbate 80 20% -ve	n/p		
							PEG 400 100% -ve	n/p	Patient is not aware of exposures to PEG 400.	
							PEG 3350 50% -ve	PEG 3350 -ve	Movicol as laxative tolerated.	
							PEG 6000 50% -ve	n/p	Patient is not aware of exposures to PEG 6000.	
5	29	F	ChAdOx-S	Allergic rhinoconjunctivitis	Level 2 Anaphylaxis Sudden onset and rapid progression of Major criteria: - Upper airway swelling (throat and uvula), tachypnoea, and increased use of accessory muscles Minor criteria: - Tachycardia, decreased level of consciousness - Difficulty breathing without wheeze or stridor	8 weeks	ChAdOx-S 10% +ve	ChAdOx-S -ve		ChAdOx-S anaphylaxis.
							Polysorbate 80 20% +ve 2% -ve	n/p		
							BNT162b 10% +ve	BNT162b -ve		BNT162b2 tolerated.
							PEG 2050 50% -ve	PEG 2050 -ve		
							PEG 400 100% -ve	n/p	Patient is not aware of exposures to PEG 400.	
							PEG 3350 50% -ve	n/p	Patient is not aware of exposures to PEG 3350.	
							PEG 6000 50% -ve	n/p	Patient is not aware of exposures to PEG 6000.	

+ve—positive result; -ve—negative result; PEG—polyethylene glycol; n/p—not performed.

**Table 3 vaccines-11-00252-t003:** Performance characteristics of skin testing and basophil activation testing.

IDT	W/V	Sensitivity		Specificity		Number of Patients
PEG 1500 or 2050	5%	100%		100%		20
Polysorbate 80	20%	100%		90%		20
BNT162b2	10%	100%		83.3%		15
ChAdOx1-S	10%	100%		100%		6
PEG 400	10%	-		65%		20
PEG 400	1%	-		90%		20
PEG 400	0.01%	-		100%		20
PEG 3350	5%	-		75%		20
PEG 3350	0.5%	-		100%		20
PEG 6000	5%	-		85%		20
PEG 6000	0.5%	-		100%		20
**BAT**		**CD63**	**CD203c**	**CD63**	**CD203c**	
PEG 1500 or 2050	10%	66.6%	33.3%	100%	66.6%	12
PEG 1500 or 2050	1%	-	0%		77.8%	12
Polysorbate 80	10%	0%	0%	100%	50%	5
Polysorbate 80	1%	-	0%	-	75%	5
BNT162b2	10%	50%	50%	88.9%	87.5%	10
BNT162b2	5%	0%	-	100%	-	10
ChAdOx1-S	10%	0%	0%	100%	75%	6
PEG 400	10%	-	-	100%	77.8%	9
PEG 400	0.1%	-	-	-	88.9%	9
PEG 3350	10%	-	-	100%	83.3%	6

## Data Availability

Data can be provided on reasonable request from the authors.

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
