# Peer review of "Skin Testing and Basophil Activation Testing Is Useful for Assessing Immediate Reactions to Polyethylene Glycol-Containing Vaccines"

_vaccines, 2023, doi:10.3390/vaccines11020252_

Round 1
Reviewer 1 Report
Li J et al. presented data regarding skin and basophil activation testing in subjects with suspected immediate allergic reaction after polyethylene glycol-containing vaccines. Authors reviewed literature concerning PEG anaphylaxis and presented 5 cases of immediate reactions after PEG and 15 cases defined as immunization stress-related responses. Skin testing with the two mRNA vaccines, PEG 2000 and polysorbate 80 resulted in good performances. BAT had high specificity. The protocol used could be tested in a more numerous cohort of subjects to validate it.
MAJOR COMMENTS:
1. The combination of literature revision and original data is quite confusing. A complete revision of PEG immediate hypersensitivity reactions has already been published before COVID vaccine cases (Wenande E et al 2016). Literature search was not indicated. I would maintain the literature data in the introduction and in the discussion without the form of a review.
2. ST and BAT in a group of non-vaccinated subjects would add more information as concern sensitivity and specificity of the test
3. The 15 non allergic patients should be better characterized (age, sex, atopy, ets…). More data should be included for the 5 allergic patients too at least for atopy
4. In the abstract authors reported the proposal to develop an algorithm for assessment of COVID-19 reactions. They referred to their applied protocol but a real algorithm is not present. Process of assessment and results are reported in Figure 1
5. Line 186: authors reported that performances at various cut-offs for ST and BAT were shown in table 3. Actually, table 3 reported sensitivity and specificity of different PEG concentrations and not cut-offs
6. Reference 14 does not report the method of basophil activation test as indicated by the authors lines 164-165. Authors should briefly describe the method or use another reference.
7. BAT for all the excipients was performed in a limited number of subjects and it is difficult to draw any conclusion.
8. In table 2, the column “challenge results” is a mix of ST results and anamnestic data, it is a bit confusing.
MINOR COMMENTS
1. Lines 34-35 “Anaphylaxis occurs rapidly and systemically, usually in systems where there are large numbers of mast cells” please add a reference for this sentence
2. Authors reported patients “who tolerated or had symptoms not consistent with anaphylaxis were classified as non-allergic”. Authors should specify if tolerated is referred to all PEG and/or vaccines.
3. In table 1, Anaphylaxis should not be under “Immunization stress-related responses”
Author Response
MAJOR COMMENTS:
- The combination of literature revision and original data is quite confusing. A complete revision of PEG immediate hypersensitivity reactions has already been published before COVID vaccine cases (Wenande E et al 2016). Literature search was not indicated. I would maintain the literature data in the introduction and in the discussion without the form of a review.
We wanted to provide an update given the previous review was pre COVID-19 and did not include PEG 2000. We have changed the section to “Introduction”.
- ST and BAT in a group of non-vaccinated subjects would add more information as concern sensitivity and specificity of the test
We have expanded the results section to identify the unvaccinated subjects in the controls. Five of the controls were unvaccinated. At the time of the study, the vaccine mandate was already rolled out in Australia so the only unvaccinated people were those pending review for a mandate exemption e.g. self-reported allergy to PEG.
Fifteen patients were found to have non-allergic presentations and were used as controls. The median age of these controls was 46 years. Thirteen of 15 patients were female. Two of 15 patients reported a history of atopy. Five patients were unvaccinated against COVID-19 at the time of review; four patients had infusion reactions to paclitaxel which responded to slowing of the infusion rate, and one patient was referred for assessment of PEG allergy but scrutiny of her current regular medications demonstrated tolerance of multiple PEG MW. Ten patients received dose 1 of a COVID-19 vaccination; 8 patients had ISRR to BNT162b and 2 to ChAdOx-S.
We also did statistical analysis using the unvaccinated controls only albeit a small number, compared to using both controls.
Performance characteristics at specified concentrations for ST and BAT are shown in Table 3. For PEG 400 and 3350, we show specificity only at serial concentrations, as we did not have allergic patients. There was no significant difference in performance characteris-tics using the 5 unvaccinated controls compared to all 15 vaccinated and unvaccinated controls in the calculation.
- The 15 non allergic patients should be better characterized (age, sex, atopy, ets…). More data should be included for the 5 allergic patients too at least for atopy
We have incorporated this in the text:
Fifteen patients were found to have non-allergic presentations and were used as controls. The median age of these controls was 46 years. Thirteen of 15 patients were female. Two of 15 patients reported a history of atopy.
- In the abstract authors reported the proposal to develop an algorithm for assessment of COVID-19 reactions. They referred to their applied protocol but a real algorithm is not present. Process of assessment and results are reported in Figure 1
We have changed this to “We performed a literature review on PEG allergy and sought to evaluate the safety and effectiveness of our protocol for assessment of COVID-19 vaccine reactions in an outpatient clinic setting.
- Line 186: authors reported that performances at various cut-offs for ST and BAT were shown in table 3. Actually, table 3 reported sensitivity and specificity of different PEG concentrations and not cut-offs
We have changed cut-offs to concentrations:
For BAT, cut-offs for up-regulation of 5% and 20% for CD63 and CD203c respectively were optimal. Performance characteristics at specified concentrations for ST and BAT are shown in Table 3.
- Reference 14 does not report the method of basophil activation test as indicated by the authors lines 164-165. Authors should briefly describe the method or use another reference.
Sorry this was in error, it has been corrected. It is reference 31.
- BAT for all the excipients was performed in a limited number of subjects and it is difficult to draw any conclusion.
The small numbers are a limitation in our study but we feel it is important to show the data we have as it may assist future research.
- In table 2, the column “challenge results” is a mix of ST results and anamnestic data, it is a bit confusing.
We have revised Table 3 substantially, hopefully it is less confusing. If there are any further suggestions for improvement, please let us know.
MINOR COMMENTS
- Lines 34-35 “Anaphylaxis occurs rapidly and systemically, usually in systems where there are large numbers of mast cells” please add a reference for this sentence
Reference 2 is now included.
- Authors reported patients “who tolerated or had symptoms not consistent with anaphylaxis were classified as non-allergic”. Authors should specify if tolerated is referred to all PEG and/or vaccines.
We have clarified this now:
Patients who tolerated or had symptoms not consistent with anaphylaxis to PEG and BNT162b and/or ChAdOx1-S were classified as non-allergic.
- In table 1, Anaphylaxis should not be under “Immunization stress-related responses”
Sorry, this was an editorial error – we have fixed it.
Reviewer 2 Report
Li et al describe 20 patients referred to their center with a possible PEG allergy. They diagnose 5 patients and proceeded with vaccination with an alternative COVID19 vaccine.
They provide an overview of their diagnostic workup and calculate sensitivity, specificity for skin tests and basophil activation testing.
The description is detailed and the methods are well-outlined (although BAT should be elaborated in more detail). However, I have several concern.
The number of cases (and controls) is limited.
The allergic cases are, as they are described at the moment, poorly convincing to be genuine anaphylaxis and PEG allergy.
- First, at least the Brighton criteria could be applied to strengthen the idea these are anaphylaxis cases. Many vaccine-induced symptoms can be regarded as subjective
- Second, the finding of objective upper airway swelling (Case 3, 4, 5) is not convincing, especially in the absence of any other mucocutaneous sign. This usually is a red herring in anaphylaxis after a parenterally administered drug. What is exactly meant (oral/pharyngeal/laryngeal/)? Many patients have been seen with presumed laryngeal edema which afterwards turned out to be ILO, some even with rash/erythema, with a preponderance of females (as in this series).
- Third, it is highly remarkable to read that all SPT with PEG, the standard in its diagnosis, were negative.
- Lastly and in line, the authors required IDT in all patients to elicit a response in their patients. Nevertheless, also ‘non-allergics’ turned out to have positive IDT. This suggests an irritative phenomenon.
IDT have been reported (repeatedly) to be associated with systemic reactions (as in this report with BNT162b2 apparently). Can the authors specify their order of testing and what timing was respected (interval between tests)?
The authors refer to their ST protocol, but this is not a protocol but merely an overview of all tested substances. It is hard to see what difference or added value this brings compared to previously published protocols (for instance, but not exclusively Banerji et al. JACI Pract PMID 333388478; Ieven et al JACI Pract PMID 3462687; Barbaud et al. PMID 3511237). An overview of the differences/added value could be relevant.
The authors mention to have performed a literature review but seem to have missed a relevant proportion of articles on this topic. For instance, the first report indicating tolerance of polysorbate in PEG allergic patients (Ieven et al JACI Pract PMID 3462687), the initial recommendation (Banerji et al. JACI Pract PMID 333388478) and current EAACI consensus (Barbaud et al. PMID 3511237) should be references.
It could be stressed that a referral for a COVID-19 mRNA vaccine related anaphylaxis should be evaluated differently. Despite billions of doses have been administered, only a handful of cases with possible PEG-allergy diagnosed after PEG-based mRNA vaccine administration have been reported.
Why did the authors not evaluate PEG 20,000?
Flow chart in figure 1 states that after evaluating the clinical history and reviewing medical notes, the authors already diagnosed 5 patients with immediate allergy to drug/vaccine containing PEG/polysorbate and diagnosing 15 with a non-allergic causes. Either the wording is essentially wrong, either the authors are able to diagnose allergy without testing. In the second round it is unclear how many were diagnosed with an allergy (i.e. after the allergy workup). Please amend to a more logic flow.
Patient details: what was the interval between event and skin testing? Were tryptase levels measured?
The authors indicate that 5% CD63 upregaulation was optimal. This is the internationally accepted cut-off for basophils, please rephrase.
It remains important to mention the responder status to control stimuli (fMLP, aIgE), which should be at least 10%, otherwise these patients should be designated as non-responders. Please also report the number of non-responders.
The authors indicate that one ‘non-allergic’ patient demonstrated tolerance of multiple PEG MW. It is important to realize that genuine PEG allergic cases can often be exposes orally to low doses of different PEG MW, whilst parenteral exposure can/will elicit reactions. The strategy to exclude PEG allergy through exposure of PEG in oral drugs (except for laxatives), is not a safe strategy and should not be promoted.
Cases (Table 2)
It could be emphasized that anaphylaxis should be treated with epinephrine and not hydrocortisone (case 1, 2)
Case 1, 2: Movicol tolerated. However in what dose? QD or as a bowel preparation? The latter is known to be an elicitor of anaphylaxis in PEG allergic patients, possibly because of the dose.
Author Response
- First, at least the Brighton criteria could be applied to strengthen the idea these are anaphylaxis cases. Many vaccine-induced symptoms can be regarded as subjective
Thank you for the suggestion. Clinicians originally graded as allergic vs non allergic using the ASCIA (Australasian Society of Clinical Immunology and Allergy) definitions https://allergy.org.au/hp/papers/acute-management-of-anaphylaxis-guidelines/ .However, we agree the Brighton criteria strengthens the idea these are anaphylaxis cases. We have included that in the methods.
Patients with history and examination consistent with anaphylaxis for a PEG MW were classified as allergic. We used the Brighton Collaboration case definition of anaphylaxis to guide our classification (30). Cases of Level 1, 2 or 3 of certainty were included as allergic.
Table 2 has been updated with how each patient met the Brighton criteria, and what level.
- Second, the finding of objective upper airway swelling (Case 3, 4, 5) is not convincing, especially in the absence of any other mucocutaneous sign. This usually is a red herring in anaphylaxis after a parenterally administered drug. What is exactly meant (oral/pharyngeal/laryngeal/)? Many patients have been seen with presumed laryngeal edema which afterwards turned out to be ILO, some even with rash/erythema, with a preponderance of females (as in this series).
Thank you for the suggestion to use the Brighton criteria - we have updated Table 2 with what the upper airway swelling exactly was, as well as correlation with the Brighton case definitions for clarity to readers as it was ambiguous previously.
- Third, it is highly remarkable to read that all SPT with PEG, the standard in its diagnosis, were negative.
This was our finding. Prior to COVID-19 pandemic, PEG allergy was mostly to PEG 3350/4000. Our study involves mainly PEG 2000 and polysorbate 80. Perhaps this is the differentiator.
- Lastly and in line, the authors required IDT in all patients to elicit a response in their patients. Nevertheless, also ‘non-allergics’ turned out to have positive IDT. This suggests an irritative phenomenon.
We believe it is because our study population was PEG 2000. Yes we agree the non allergics with positive IDT suggest an irritative phenomenon. We think it’s important for readers to know that we found IDT irritative at these concentrations for PEG MW.
IDT have been reported (repeatedly) to be associated with systemic reactions (as in this report with BNT162b2 apparently). Can the authors specify their order of testing and what timing was respected (interval between tests)?
For the vaccines, we performed SPT first, then waited 20 minutes, then IDT at 1%, if negative, then 20 minutes later, IDT at 10%.
For PEG 400, we performed SPT first, then waited 20 minutes, then IDT at 1%, if negative, then 20 minutes later, IDT at 10%, if negative then IDT at 100%.
For all other PEG MW, we performed SPT first, then waited 20 minutes, then IDT at 0.5%, if negative, then 20 minutes later, IDT at 5%, if negative then IDT at 50%.
We have improved Figure 1 to incorporate this point.
The authors refer to their ST protocol, but this is not a protocol but merely an overview of all tested substances. It is hard to see what difference or added value this brings compared to previously published protocols (for instance, but not exclusively Banerji et al. JACI Pract PMID 333388478; Ieven et al JACI Pract PMID 3462687; Barbaud et al. PMID 3511237). An overview of the differences/added value could be relevant.
Adding to the existing literature (25-27), we provide information on the safety of IDT using pure PEG at specified concentrations in the outpatient setting, as well as provide some guidance on irritant concentrations for IDT. We provide an outpatient-friendly protocol using vaccines and pure PEG only to simplify ST.
The authors mention to have performed a literature review but seem to have missed a relevant proportion of articles on this topic. For instance, the first report indicating tolerance of polysorbate in PEG allergic patients (Ieven et al JACI Pract PMID 3462687), the initial recommendation (Banerji et al. JACI Pract PMID 333388478) and current EAACI consensus (Barbaud et al. PMID 3511237) should be references.
We have included all this in the current version.
Skin testing (ST), comprising skin prick testing (SPT), and intradermal testing (IDT), and basophil activation testing (BAT) are used to aid diagnosis of drug and vaccine ana-phylaxis including PEG anaphylaxis. ST generally comprises SPT at first instance, as sys-temic reactions from SPT are well described in PEG allergic patients, however escalation to IDT is possible after negative SPT (11). ST with higher PEG MW is more likely to result in systemic reaction in allergic patients (11). Recommendations have continued to evolve and there remains no standardized method. In 2020, shortly after COVID-19 vaccinations became available, Banerji et al provided guidance in patients with allergy or atopy for first time COVID-19 vaccine administration (25). The authors suggested ST for high risk pa-tients, comprising those with suspected PEG anaphylaxis. ST was performed as SPT to PEG 3350, polysorbate 80 and polysorbate 20 containing medications, and if negative, followed by IDT (25). In 2021, Ieven et al identified PEG allergic patients, predominantly to PEG 3500/4000 but also PEG 400 and PEG 6000, who were able to tolerate polysorbate-80 and polysorbate-80 containing vaccines by ST (26). This was performed as SPT to a range of PEG MW in pure form and PEG containing medications, and to polysorbate-80 in pure form undiluted (26). IDT was only performed to PEG 3350 and polysorbate 80 if SPT was negative (26). In 2022, the European Association of Allergy & Immunology (EAACI) iden-tified high risk groups for COVID-19 vaccination, comprising those with immediate reac-tions (within 2 hours of administration) to COVID-19 vaccines, or anaphylaxis; and those with a history of suspected PEG anaphylaxis (27). It was recommended that these patients are referred for allergy assessment with SPT to COVID-19 vaccine in neat concentration, and PEG and polysorbate 80 in either pure form at up to 50% dilution in water, or as PEG-containing medications (27). Further guidance for SPT testing concentrations to pure PEGwas described by Bruusgaard-Mouritsen et al in 2022, the authors suggested using PEG 300 at 100% weight/volume (w/v), PEG 3000 at 50% w/v, PEG 6000 at 50% w/v, and polysorbate 80 at 20% w/v (28). SPT to PEG 20,000 was recommended to be performed in a stepwise fashion with 0.01%, 0.1%, 1%, 10%, and 20% w/v until a positive response was reached. The protocol, however, did not include PEG 2000. In addition to testing the cul-prit PEG MW, it was considered that ST with MW higher than the culprit MW may im-prove ST sensitivity, and should be performed prior to excluding PEG allergy (11).
It could be stressed that a referral for a COVID-19 mRNA vaccine related anaphylaxis should be evaluated differently. Despite billions of doses have been administered, only a handful of cases with possible PEG-allergy diagnosed after PEG-based mRNA vaccine administration have been reported.
We have now done this in the abstract, methods and discussion section.
Why did the authors not evaluate PEG 20,000?
We have now included the reason why in the methods. We performed our testing in outpatient clinic rooms.
PEG with MW above 6000g/mol were not utilized as the outpatient clinic risk matrix did not permit use due to high risk of systemic reaction.
Flow chart in figure 1 states that after evaluating the clinical history and reviewing medical notes, the authors already diagnosed 5 patients with immediate allergy to drug/vaccine containing PEG/polysorbate and diagnosing 15 with a non-allergic causes. Either the wording is essentially wrong, either the authors are able to diagnose allergy without testing. In the second round it is unclear how many were diagnosed with an allergy (i.e. after the allergy workup). Please amend to a more logic flow.
We have clarified the wording in Figure 1 to avoid confusion.
Most pertinently, “5 patients with immediate allergy” has been changed to “5 patients meeting Brighton Case Definition for Anaphylaxis…”, and “Allergy workup” changed to “Skin and BAT testing”
Patient details: what was the interval between event and skin testing? Were tryptase levels measured?
The interval is listed in Table 2 – under Time from reaction to assessment. Tryptase levels were unfortunately not available, whether tryptase levels are taken in the Emergency Department in Australia is variable and often not done.
The authors indicate that 5% CD63 upregaulation was optimal. This is the internationally accepted cut-off for basophils, please rephrase.
We have rephrased simply as “For BAT, cut-offs for up-regulation of 5% and 20% for CD63 and CD203c respectively were used.”
It remains important to mention the responder status to control stimuli (fMLP, aIgE), which should be at least 10%, otherwise these patients should be designated as non-responders. Please also report the number of non-responders.
We had no non-responders, we would have indicated not have provided the BAT result for non responders.
The authors indicate that one ‘non-allergic’ patient demonstrated tolerance of multiple PEG MW. It is important to realize that genuine PEG allergic cases can often be exposes orally to low doses of different PEG MW, whilst parenteral exposure can/will elicit reactions. The strategy to exclude PEG allergy through exposure of PEG in oral drugs (except for laxatives), is not a safe strategy and should not be promoted.
We have clarified Table 2 using two columns – “history of exposure to PEG” and “outcome of COVID-19 vaccination” to reduce confusion especially in relation to your last point that exposure to oral PEG to exclude PEG allergy is not safe. We merely indicate “Movicol and Coloxyl tolerated as laxative” etc to avoid this.
Cases (Table 2)
It could be emphasized that anaphylaxis should be treated with epinephrine and not hydrocortisone (case 1, 2)
We have taken on board your comments and overhauled Table 2 with a new column Brighton Collaboration Case Definition for Anaphylaxis and removed references to hydrocortisone (which is unfortunately still frequently used in management of anaphylaxis).
Case 1, 2: Movicol tolerated. However in what dose? QD or as a bowel preparation? The latter is known to be an elicitor of anaphylaxis in PEG allergic patients, possibly because of the dose.
We have clarified this in Table 2, e.g. “Movicol tolerated as laxative”.
Round 2
Reviewer 1 Report
Line 171: “All patient reviews and drug allergy testing was performed”, please change was with were
Basophil activation test method was referred to citation 31, however indication when BAT is considered positive (% of CD63 or CD203c and stimulation index higher than?) should be included in the method section.
Author Response
Thank you, we have made the relevant changes in the manuscript text, highlighted in red.